# Surface-Modified In_2_O_3_ for High-Throughput Screening of Volatile Gas Sensors in Diesel and Gasoline

**DOI:** 10.3390/ma16041517

**Published:** 2023-02-11

**Authors:** Deqi Zhang, Shenghui Guo, Jiyun Gao, Li Yang, Ye Zhu, Yanjia Ma, Ming Hou

**Affiliations:** 1State Key Laboratory of Complex Nonferrous Metal Resources Clean Utilization, Kunming University of Science and Technology, Kunming 650093, China; 2Faculty of Metallurgical and Energy Engineering, Kunming University of Science and Technology, Kunming 650093, China

**Keywords:** high-throughput screening, surface modification, In_2_O_3_, gas sensor, fuel volatiles

## Abstract

In this paper, with the help of the method of composite materials science, parallel synthesis and high-throughput screening were used to prepare gas sensors with different molar ratios of rare earths and precious metals modified In_2_O_3_, which could be used to monitor and warn the early leakage of gasoline and diesel. Through high-throughput screening, it is found that the effect of rare earth metal modification on gas sensitivity improvement is better than other metals, especially 0.5 mol% Gd modified In_2_O_3_ (Gd_0.5_In) gas sensor has a high response to 100 ppm gasoline (*R_a_*/*R_g_* = 6.1) and diesel (*R_a_*/*R_g_* = 5) volatiles at 250 °C. Compared with the existing literature, the sensor has low detection concentration and suitable stability. This is mainly due to the alteration of surface chemisorption oxygen caused by the catalysis and modification of rare earth itself.

## 1. Introduction

With the rapid development of industry, energy demand is also increasing sharply. In 2021, the average daily consumption of oil in the United States was about 19.78 million barrels, about 8% higher than that in 2020 [1], among which gasoline and diesel are the most consumed petroleum products. The volatile gas produced by the inhalation of gasoline and diesel will cause dizziness, nausea, and limb weakness [2,3,4], and when the concentration of volatile gasoline and diesel reaches the low flammability limit (LELs), safety risks will occur in the case of high temperature or ignition source [5]. Therefore, it is significant for human health and safety products to develop gas sensors for volatile compounds of gasoline and diesel.

Metal oxide semiconductor (MOS) has become one of the most studied and widely used gas-sensitive materials due to its advantages of high sensitivity, low price, easy integration, and accessible collection and analysis of measurement signals [6,7,8]. The sensing mechanism is the contact between gas molecules and metal oxide semiconductors, the exchange of electrons on the semiconductor surface, and the absorption and desorption of gas, which will cause a specific change in the semiconductor resistance. Indium oxide (In_2_O_3_) is a typical n-type metal oxide semiconductor with high conductivity, low resistivity, and a wide band gap (3.55~3.75 eV), making it an excellent gas-sensitive material [9,10]. However, there are still shortcomings of low sensitivity and high detection concentration in the detection of volatile substances of gasoline and diesel, which cannot play an early warning role in the leakage of volatile substances of low concentration of gasoline and diesel [11,12]. Surface modification has been proven to be an effective and simple method to improve the sensing performance of metal oxide gas-sensitive materials [13,14,15]. By using the activity of rare earths and precious metals, the surface states of In_2_O_3_ gas-sensitive materials can be changed to provide more active sites for gas-sensitive reactions, and at the same time, it is beneficial to form lattice distortion and increase oxygen vacancy on the surface of In_2_O_3_ gas-sensitive materials. However, the application of gas sensors based on In_2_O_3_ nanoparticles modified with rare earths and precious metals surface to the detection of volatile compounds in diesel and gasoline is rarely reported.

Due to the diversity of surface modification types, the efficiency of screening high-performance gas-sensitive materials by traditional methods cannot meet the needs of society. Therefore, the method of combinatorial materials science [16] has been widely concerned as an efficient method. Combinatorial materials science mainly consists of parallel synthesis technology and high-throughput characterization technology [17]. Parallel synthesis refers to the simultaneous synthesis of a large number of material samples, which can significantly improve the efficiency of material synthesis and avoid human errors in the process of material synthesis. High-throughput characterization technology is to simultaneously characterize a large number of materials, to improve the efficiency of high-performance material screening.

In this paper, a sensor for volatile compounds of gasoline and diesel was prepared based on rare earths and precious metals modified by nano In_2_O_3_ particles by means of composite materials science. The results show that the rare earths and precious metals modification can effectively enhance the response performance of In_2_O_3_ to the volatiles of gasoline and diesel. The sensor based on Gd_0.5_In has a high response to the volatiles of gasoline and diesel at 250 °C, with reliable stability and repeatability.

## 2. Experimental

### 2.1. Raw Material Preparation

The chemicals used in this work were purchased from Aladdin Chemical Group Co., LTD., China, and are analytical grade, requiring no further purification. Dissolve 0.50 mmol In (NO_3_)_3_·5H_2_O in 20 mL DMF and stir to form a uniform, transparent solution. Then, 0.01 mol urea was slowly added into the solution while stirring and continued stirring for 1 h until a transparent solution was formed. The solution was then transferred to a 25 mL reactor and heated at 100 °C for 24 h. After cooling, the white precipitates were centrifuged with deionized water and ethanol 3 times each. After drying for 2 h in the air at 60 °C and calcination for 2 h at 500 °C, In_2_O_3_ nanoparticle powder was obtained.

### 2.2. Gas-Sensitive Film Preparation and Device Packaging

The types of surface modification elements and the surface modification mole are shown in Table 1, with a concentration of 0.01 g/mL. In this experiment, a parallel gas-sensitive film synthesizer (Figure 1a) was used to prepare In_2_O_3_ gas-sensitive films modified with metal ions. In_2_O_3_ (can be sprayed raw material liquid) dispersion by ball mill. Then, the premixing function of the platform is used to set the amount of rare earths and precious metals solution and In_2_O_3_ raw material solution according to the modification ratio. According to the setting, the device automatically completes the preparation of gas sensing materials (i.e., premixed solution) of rare earths and precious metals modified In_2_O_3_ with different molar ratios. Then, the premixed solution was deposited on the prepared substrate through the transfer function of the platform [18]. The substrate with a gas-sensitive film was first annealed at 350 °C for 2 h to remove organic matter and then annealed at 550 °C for 2 h to denser the film. Finally, a substrate with a gas-sensitive film is made into a sensor. The prepared sensor was named; for example, the sensor with 0.1 mol% Ce modified In_2_O_3_ was named Ce_0_._1_In, in which the unmodified In_2_O_3_ sensor was the blank control sample.

### 2.3. Gas Sensitivity Test

In this experiment, the constant temperature gas-sensitive properties of 61 kinds of gas-sensitive materials modified with In_2_O_3_-based surface were tested in constant temperature mode at temperature points between 200 °C and 400 °C every 50 °C. The test gas was configured with the headspace method. Before gas distribution, dry air was used to clean the stainless-steel gas cylinder heated to 120 °C several times. Then according to the gas concentration to be configured, the ideal gas PV = nRT formula is used to calculate the corresponding solution volume. Then use the pipette gun to take the corresponding amount into the valve chamber and heat it to volatilize, and finally, use dry air to drive the gas in the valve chamber into the cylinder pressure required.

To simultaneously test the gas-sensitive properties of 61 gas-sensitive materials, an unmanned high-throughput gas-sensitive performance tester (Figure 1b) was used in this paper to test the gas-sensitive properties. Firstly, the optimal operating temperature of the gas sensor was determined, and the steady-state response of the 75 ppm gasoline and diesel volatiles at 200, 250, 300, 350, and 400 °C was tested under the air back. After determining the optimal temperature, the steady-state response of gasoline and diesel at different concentrations of 30, 50, 70, and 100 ppm was tested. Each set of tests was repeated three times. The response value is defined by the ratio between the steady-state resistance value *R_a_* in dry air and the steady-state resistance value *R_g_* in the test atmosphere, that is, *S* = *R_a_*/*R_g_*. The response recovery time is the time when the gas-sensitive material adsorbed and desorbed the target gas to 90% of the stable value.

### 2.4. Characterization

The phase analysis of In_2_O_3_ nano-powder was carried out by X-ray diffractometer (XRD, Rigaku D/max-2500). The parameters were as follows: copper target K line (wavelength 0.15406 nm), scanning angle 2θ = 10°–90°. The particle size of In_2_O_3_ powder, the surface morphology, and the cross-section morphology of Pr_0.5_In gas-sensitive film were analyzed, and the surface element distribution of Sb_0.5_In was analyzed by field emission scanning electron microscope (FESEM, JSM-7100F, JEOL). 

## 3. Results and Discussion

### 3.1. Material Structure and Morphology Characterization

Figure 2 is the characterization result of nano In_2_O_3_ powder, and Figure 2a is the XRD pattern of nano In_2_O_3_ powder. XRD peaks can be seen to be consistent with the standard XRD pattern of cubic phase In_2_O_3_ (JCPDS NO.65-3170), indicating that the phase of the synthetic material is correct. Figure 2b is the FESEM picture of the In_2_O_3_ nano-powder. It can be seen that it is a uniformly dispersed nanoparticle with a particle size of about 30 nm. 

Figure 3 shows the parallel film forming results of the high flux gas-sensitive film parallel synthesis instrument. Figure 3a shows the Pr_0.5_In-sensitive film prepared by the high flux gas-sensitive film parallel synthesis instrument. It can be seen that the gas-sensitive film is approximately circular, and the surface is flat without macroscopic and microscopic cracks. However, some clumps exist, which may be due to the agglomeration of the raw materials themselves or the lack of appropriate additives in the stream premix, resulting in phagocytic clumps between the grains during the film sintering. Figure 3b shows the SEM and EDS scanning images of Sb_0.5_In gas-sensitive film, and it can be seen that In, O, and Sb are uniformly distributed.

### 3.2. Gas-Sensitive Characteristics

The sensing performance of the gas sensor is closely related to the operating temperature. To determine the optimal operating temperature, we performed a high-throughput screening of 61 prepared sensors for gas sensitivity tests of 50 ppm gasoline and diesel volatiles at 200 °C to 400 °C. The results are shown in Figure 4. The abscissa represents different modification elements, and the ordinate represents different test temperatures. The intersection points of the horizontal and vertical coordinates in the figure represent the response values of modification elements of 0.1–0.5% in this temperature from front to back, respectively. In Figure 4a, the test conditions of the x-coordinate modified element Ru and the y-coordinate 200 °C are taken as an example. The intersection points of the coordinate represent the response values of the sensor modified with In_2_O_3_ of 0.1%Ru, 0.2%Ru, 0.3%Ru, 0.4%Ru, and 0.5%Ru to gasoline volatiles of 50 ppm at 200 °C, respectively. All sensors increase in response value with increasing temperature. This is because the response value increases as the temperature increases, providing more energy to activate the reacting molecules [19]. With the increase in the modification amount of precious metals, the response of some sensors to the volatiles of gasoline and diesel is weakened. This is due to the surface disorder caused by excessive surface modification amount, which leads to the increase in the density of the surface states of gas-sensitive materials, resulting in the binding of the Fermi level on the surface of gas-sensitive materials and the reduction of the response of gas sensors [20]. At the same time, due to the small difference in response values under different temperatures, considering the device maintenance cost and practical application scenarios, the optimal temperature of the sensor to the volatile matter of gasoline and diesel was determined to be 250 °C. 

As shown in Figure 5 (The intersection points of the horizontal and vertical coordinates in the figure represent the response values of the test concentration of the modification elements representing 0.1–0.5% of the modification amount from front to back), the response performance of the prepared sensors to the volatiles of gasoline and diesel at 30, 50, 70, and 100 ppm were respectively tested at the optimum operating temperature. The response value of all the sensors increases with the concentration of the test gas. Through high-throughput screening, it was found that the effect of rare earth metal modification on gas sensitivity was better than that of precious metals, and Gd_0.5_In was the most sensitive sensor to gasoline volatiles and diesel volatiles at 100 ppm. The Gd_0.5_In response to 100 ppm gasoline volatiles and diesel volatiles is 6.1 and 5, respectively, which are 3.1 times and 2.5 times of unmodified In_2_O_3_ (*R_a_*/*R_g_* = 2), respectively. It is verified that surface modification is a way to improve the sensing performance of metal oxide gas-sensitive materials.

Next, we focused on the gas-sensitive performance of Gd_0.5_In screened with high throughput. Figure 6 is the function-fitting diagram of gasoline and diesel volatiles with Gd_0.5_In at 250 °C for 30–100 ppm, respectively. Among them, the correlation coefficient R^2^ > 0.995 for gasoline volatiles in the range of 30–100 ppm (Figure 6), and the correlation coefficient R^2^ > 0.952 for diesel volatiles in the range of 30–100 ppm (Figure 6). The results show that the 0.5 mol% Gd modified In_2_O_3_ sensor has an excellent linear relationship with the volatile concentration of gasoline and diesel [21]. Figure 7 shows the response recovery characteristics of Gd_0.5_In to 100 ppm gasoline and diesel volatiles at 250 °C. It can be seen that with the input of gasoline and diesel volatiles, the resistance of Gd_0.5_In rapidly decreases and becomes stable, and when the input of gasoline and diesel volatiles is stopped, the resistance value increases rapidly and returns to the initial resistance. The results show that the response time and recovery time of Gd_0.5_In to gasoline volatiles are 184.6 s and 196.6 s when 100 ppm of target gas is tested at 250 °C. The response time and recovery time of Gd_0.5_In to diesel volatiles are 180.4 s and 195.8 s.

To explore the influence of the sensor selectivity under a complex environment, the response performance of Gd_0_._5_In to methanol, ethanol, formaldehyde, benzene, acetone, hydrogen, and methane under the same conditions was also studied. The hydrogen and methane were purchased from China Kunming Pengyida Gas Products Co., LTD., and the other gases were prepared by the headspace method. Figure 8 shows the response histogram of the target gas at 250 °C to 100 ppm. The response values of Gd_0.5_In to the volatiles of gasoline and diesel are significantly higher than those of other gases, among which the response values of ethanol and benzene are 3.6 and 1.3, respectively, indicating that the sensitivity of Gd_0.5_In to the volatiles of gasoline is 1.7 times and 4.8 times higher than that of ethanol and benzene. The sensitivity of diesel volatile compounds is 1.4 times and 3.8 times higher than that of ethanol and benzene.

The stability of Gd_0.5_In is tested. Figure 9a shows the reversible cycle curve of three response processes continuously tested. The original response value is maintained without a significant decrease, indicating that the Gd_0.5_In sensor has stable and repeatable response characteristics. Figure 9b shows the long-term stability of the device tested at 250 °C for 50 days. It can be seen that the response value of Gd_0.5_In to the volatiles of gasoline and diesel maintains an absolute deviation of 17% and 12% fluctuation within 50 days, indicating that the sensor has suitable stability. Table 2 shows a comparison of the gas-sensitive performance of our sensor with some previously reported gasoline and diesel sensors. The comparison shows that our prepared sensor has a lower detection concentration (100 ppm), a higher response value, and a shorter corresponding recovery time for gasoline and diesel volatiles, which is fully adequate for the detection and early warning role of gasoline and diesel early leaks.

### 3.3. Explanation of the Gas-Sensitive Mechanism

According to the above results, different modification elements have different effects, and gas-sensitive materials modified with the same element have different effects on different gases. Among them, the effect of rare earth metal modification on gas sensitivity is better than that of alkali metals and precious metals. The gas-sensitive mechanism of metal oxide gas-sensitive materials is mainly the oxygen ionization model and oxygen vacancy model [22,23,24]. As we all know, In_2_O_3_ is an n-type metal oxide gas-sensitive material. When In_2_O_3_ is in contact with the reducing gas, due to the high activity of the gas molecules, when the gas is adsorbed on the surface of the material and in the pore, and there is a chemical reaction between the effective sites on the material, resulting in chemical adsorption, the electrons in the gas molecules will be transferred under the adsorption of oxygen ions on the material, causing electrons to accumulate on the surface of the material and the surface of the pore. At this time, the original electron adsorbing oxygen ion will release the raw material crystal electron back in the process of electron adsorption, which makes the material electron transfer, as shown by the increase in conductance and the decrease in resistance. On the test system, the resistance appears to be smaller than the initial state. When the material is in complete contact with the gas, the reaction sites on the surface of the material are saturated with adsorption, and when the electron transfer and adsorption reach equilibrium, the conductance of each reaction site does not change, and the resistance value is stable. After the material is removed from the test gas, adsorptive molecules at each reaction site will undergo desorption. At the end of the reaction, the electrons in the crystal will gather and combine again due to the oxygen ions on the surface of the material, resulting in the reduction of free electrons in the crystal and the reduction of conductance, which is manifested as the rise of resistance and return to the initial resistance state. During the gas reaction, the adsorbed oxygen and the sensitive gas undergo a redox reaction, and the change in the adsorbed oxygen will cause a change in the resistance. Compared with the gas-sensitive materials modified with alkali and precious metals, the gas-sensitive properties of the materials modified with rare earth metal ions improved more significantly. This is related to the unique catalytic ability and electronic activity of rare earth metals [25]. At present, the sensitization mechanism of rare earth metal catalysts is mainly divided into two parts [26]: chemical and electronic sensitization. For chemical sensitization, the reaction between gas and semiconductor can be improved by the catalysis of rare earth. The gas molecules preferentially decompose on the surface of the rare earth metal catalyst, and the products then move to the semiconductor surface. For electron sensitization, a change in the chemical state of the rare earth metal results in a change in the electronic state of the semiconductor.

## 4. Conclusions

In this paper, a combinatorial materials science approach was used to prepare gas-sensitive materials with different rare earths and precious metals modified with In_2_O_3_ using In_2_O_3_ nanoparticles as a substrate by parallel synthesis techniques. High-throughput screening of gasoline and diesel volatiles revealed that the modification of rare earth metals improved the gas sensitivity better than other modified metals. In particular, the Gd_0.5_In gas sensor showed a high response to 100 ppm gasoline (*R_a_*/*R_g_* = 6.1) and diesel (*R_a_*/*R_g_* = 5) volatiles at 250 °C, which was 3.1 and 2.5 times higher than that of unmodified In_2_O_3_ (*R_a_*/*R_g_* = 2). In comparison with the existing literature, the Gd_0.5_In gas sensor prepared here has a better response and detects lower concentrations. It is important for monitoring and early warning of early leaks of gasoline and diesel in product refining and terminal production.

## Figures and Tables

**Figure 1 materials-16-01517-f001:**
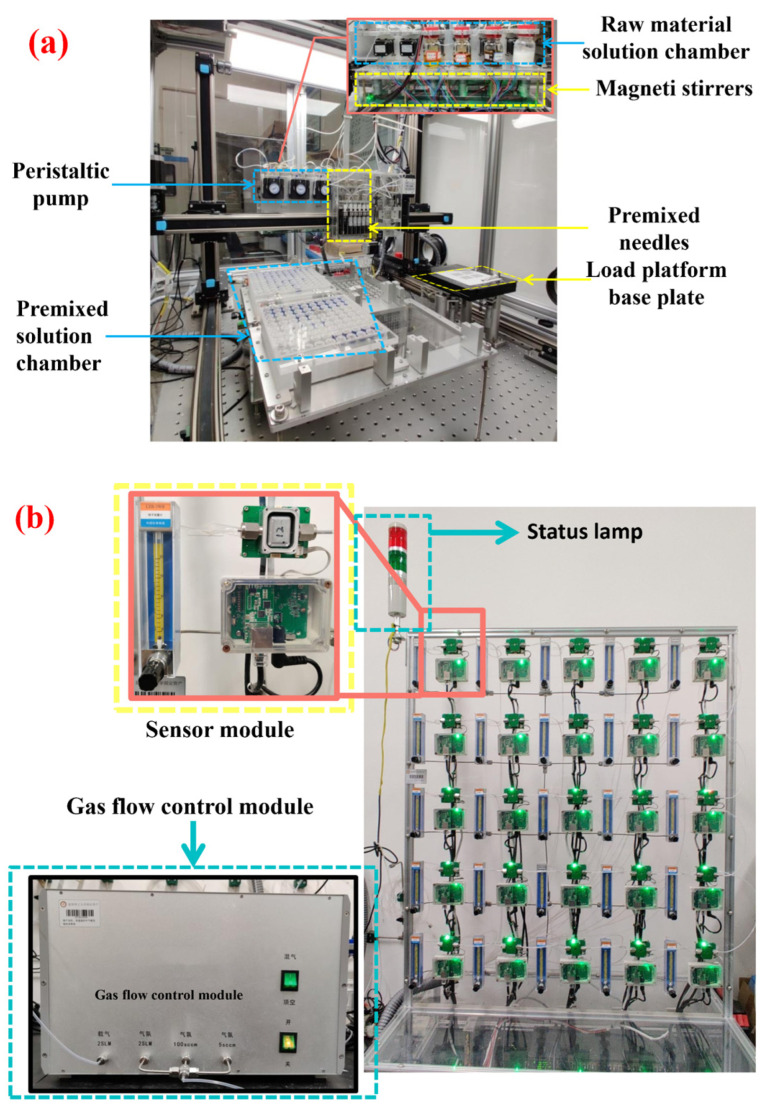
Physical picture of equipment. (**a**) Parallel gas-sensitive film synthesizer; (**b**) high-throughput gas-sensitive performance tester.

**Figure 2 materials-16-01517-f002:**
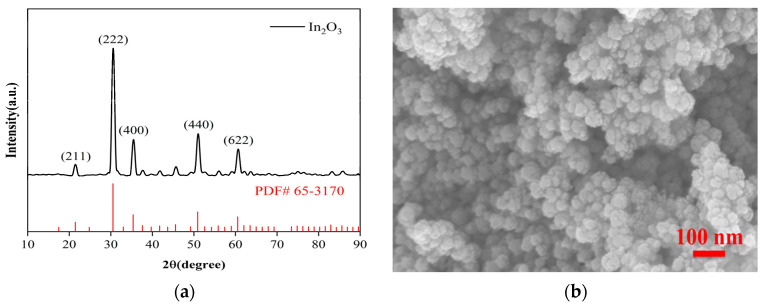
In_2_O_3_ nanometer powder characterization picture. (**a**) XRD pattern of In_2_O_3_; (**b**) FESEM picture of In_2_O_3_ nano-powder.

**Figure 3 materials-16-01517-f003:**
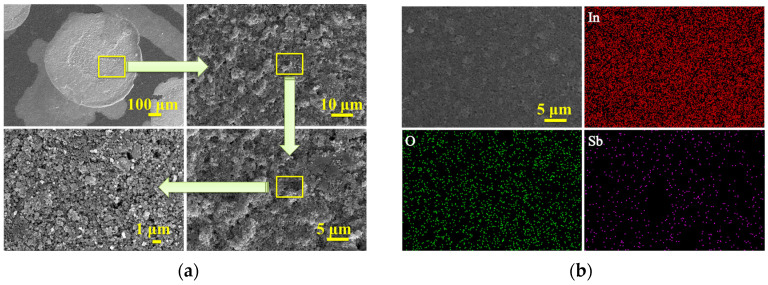
(**a**) Macroscopic and microscopic FESEM diagrams of Pr_0_._5_In gas-sensitive film; (**b**) EDS diagram of Sb_0.5_In gas-sensitive film.

**Figure 4 materials-16-01517-f004:**
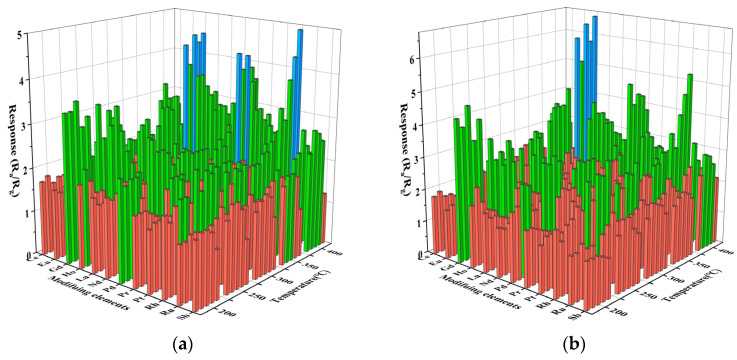
Characteristics of surface-modified In_2_O_3_ gas sensor for 50 ppm gasoline and diesel volatile compounds at 200–400 °C. (**a**) Gasoline; (**b**) diesel.

**Figure 5 materials-16-01517-f005:**
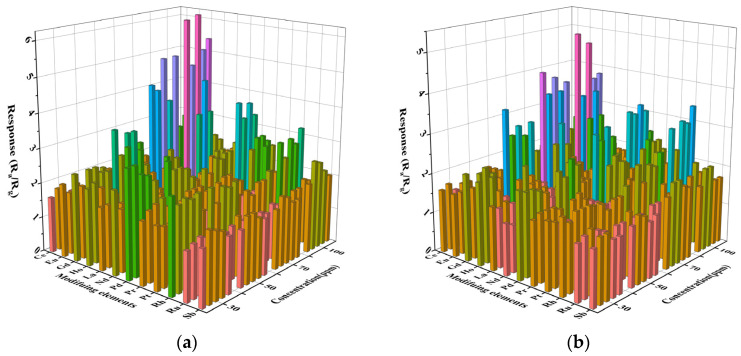
Characteristics of surface-modified In_2_O_3_ gas sensor for the determination of volatile compounds in gasoline and diesel at different concentrations at 250 °C. (**a**) Gasoline; (**b**) Diesel.

**Figure 6 materials-16-01517-f006:**
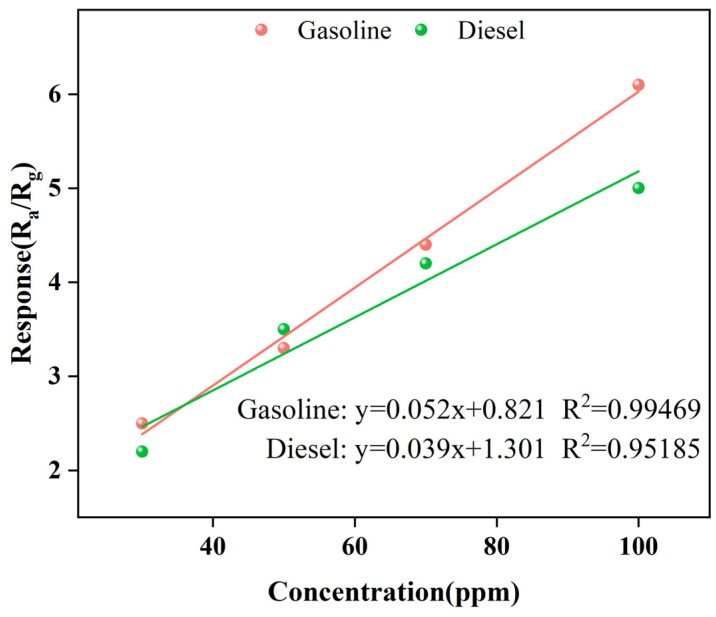
Function-fitting diagram of gasoline and diesel volatiles at different concentrations by Gd_0.5_In gas sensor at 250 °C.

**Figure 7 materials-16-01517-f007:**
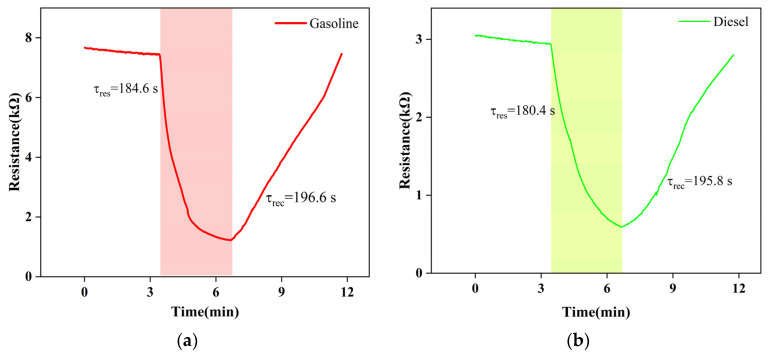
Recovery characteristics of Gd_0.5_In response to 100 ppm gasoline and diesel volatiles at 250 °C. (**a**) Gasoline; (**b**) diesel.

**Figure 8 materials-16-01517-f008:**
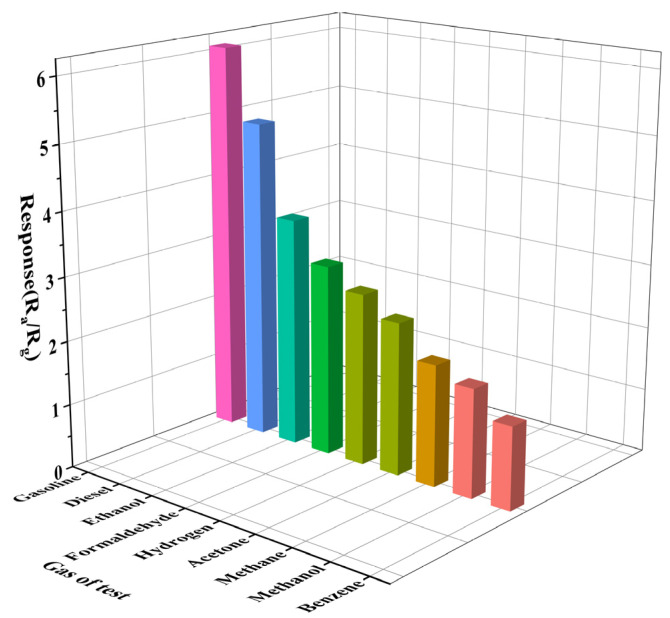
Response of Gd_0.5_In to 100 ppm of different test gases at 250 °C.

**Figure 9 materials-16-01517-f009:**
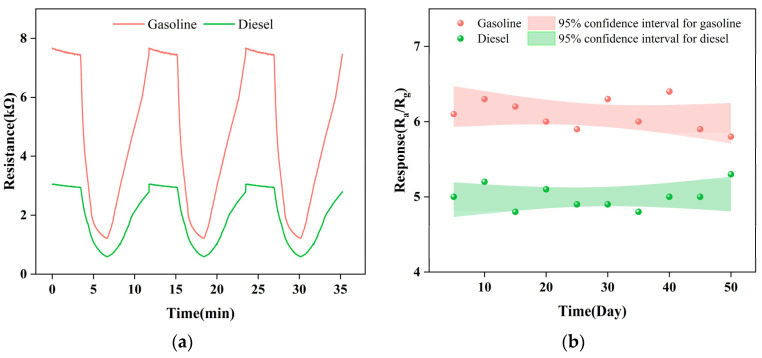
Repeatability and long-term stability of Gd_0.5_In gas sensors for gasoline and diesel volatiles. (**a**) Continuously test the reversible cyclic curves of the three response processes; (**b**) long-term stability tests within 50 days.

**Table 1 materials-16-01517-t001:** Modifies the type of element and its added proportions.

Modifying Elements	Additive	Surface Mole Ratio (%)
Ce	CeCl_3_·7H_2_O	0.1, 0.2, 0.3, 0.4, 0.5
Eu	EuCl_3_·6H_2_O	
Gd	Gd(NO_3_)_3_	
Ho	HoCl_3_·6H_2_O	
La	LaCl_3_·6H_2_O	
Nd	Nd(NO_3_)_3_·6H_2_O	
Pd	PdCl_2_	
Pr	Pr(NO_3_)_3_	
Pt	H_2_PtCl_6_·6H_2_O	
Rh	RhCl_3_·3H_2_O	
Ru	RuCl_3_·3H_2_O	
Sb	SbCl_3_	

**Table 2 materials-16-01517-t002:** Compared with the properties of volatile sensing materials for gasoline and diesel reported in previous literature.

Sensing Materials	Gas of Test	T (°C)	C (ppm)	Response	τ_res_/τ_rec_ (s)	Refs.
Pt-SnO_2_	Diesel	250	Volatile gas	2.85	314/309	[12]
Mg-In_2_O_3_	Diesel	225	Volatile gas	2.07	299/663	[12]
Pt-ZnO	Diesel	250	Volatile gas	2.59	335/698	[12]
SnO_2_ nanomaterials	Gasoline	300	200	6.1	-	[11]
Gd_0.5_In	Diesel	250	100	5	184.6/196.6	This work
Gd_0.5_In	Gasoline	250	100	6.1	180.4/195.8	

## Data Availability

Not applicable.

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
