# Peer review of "Surface-Modified In2O3 for High-Throughput Screening of Volatile Gas Sensors in Diesel and Gasoline"

_materials, 2023, doi:10.3390/ma16041517_

Round 1

Reviewer 1 Report

Zhang et al. have presented the manuscript titled: Surface-modified In2O3 for high throughput screening of volatile gas sensors in diesel and gasoline. Overall presentation of the article is good, but there require many modification before being publish, suggestions are as follow;

1.       In abstract, first two sentences are making no sense at all; there are many grammatical and technical language mistakes.

2.      Abstract, line 18, “Low detection concentration and good stability.” Can authors tell what this sentence states?

3.      In introduction, “In 2021, the average daily consumption of oil in the United States is about 19.78 million barrels”, please use past tense.

4.      In the method section, add the purity and manufacturer of the chemicals. Why the additives Gd, Sb and Pr are not discussed in method section?

5.      XRD is the most important tool for the confirmation of the successful fabrication of the material. I was wondering why authors have not added the XRD patterns for the In2O3-rare earth added samples? As 0.4 or 0.5 are high ratios they must have the influence on the parent structure of In2O3.

6.      In the SEM analysis of Figure 2b and 3a, there exist so many holes, which can influence the properties of the materials, Have authors analyzed the porosity of the samples?

7.      As authors have provided the elemental mapping, they must have also measured the EDX for this, it will be interesting to observe the prefabrication and post fabricating ratios of the constituents.

8.      In the section 3.2, authors claim the measurement of Gas-sensitive characteristics. For which sample have they made these observations presented in Figure 4? i.e., in Figure 4, which color is representing which sample?

9.      Same as point number 8, what samples are used for Figure 5?

10.  For gas sensitivity measurements, why authors only focused on Gd0.5In, sample and ignored other samples?

11.  Overall, I suggest the authors to revise whole manuscript carefully, as there are many subscript and superscript mistakes.

Reviewer 2 Report

As presented, the relationship between the Experimental and Results and discussion sections is difficult to follow. Example: what about the correlation between table 1 (various modifying elements, surface molar ratio, etc.) and other part of the manuscript? In this regard, please make the main text concise and make readers understand this manuscript effectively. Results and discussion section must be considerably improved/ more technically presented. More care should be taken to present the results and facilitate understanding of the work. Please reorganize/ reconsider the manuscript in order to highlight the most significant and unexpected results, identify correlations, patterns and relationships among the data, speculations, limitations of work and deductive arguments.

Therefore, I consider that the paper is not proper for publication in the present format in the Materials journal and must be "Reject". Nevertheless, the efforts of performing all the experiments have been significant and I hope that in the near future all the issues will be solved.

Reviewer 3 Report

The manuscript entitled "Surface-modified In2O3 for high throughput screening of volatile gas sensors in diesel and gasoline" demonstrates the gas sensitivity can be improved with the surface modified Indium Oxide(In2O3)sensor according to the high throughput screening result, which indicates a potential to product the early warning device to reduce the damage to human and environment.

1. Abstract should be polished. It lacks some main content and cannot arouse readers' interest.

2. Can the authors show the standard XRD result for the In2O3  powder and overlap it with the experiment result?

3. Section 3.2, the first paragraph should be polished to elucidate why the response value co-increase with the temperature. Also, it's noticed there are vibrations and decreases during 200 to 400. Can the authors elucidate it?

4. Most of the citations are old. Can the authors include more recent publications to show the current progress in the field.

Minor Comments:

Figure 2, the label (a) (b) should be the same height and same character size.

Figure3,  the scale bar is hard to be recognized. And don't need (a) (b) again label for the inset figures.

Round 2

Reviewer 1 Report

Authors have revised the manuscript very well according to my suggestions, I think XRD and EDS would make it more promising for the readers. I suggest the this manuscript to be published in present form.

Author Response

Response to Reviewer 1 Comments

Authors have revised the manuscript very well according to my suggestions, I think XRD and EDS would make it more promising for the readers. I suggest the this manuscript to be published in present form.

Response: Thank you for taking the time to process our submission entitled "Surface-modified In2O3 for high throughput screening of volatile gas sensors in diesel and gasoline" original paper. Thank you for your careful review and affirmation of our work. Your comments were highly insightful and enabled us to greatly improve the quality of our manuscript.

Reviewer 2 Report

Based on this revised version of the manuscript, unfortunately there are still unanswered questions and weaknesses in this paper. Please carefully revise the manuscript according to my previous suggestions.

The discussion presented is poor, in terms of discussing its results and comparing them with the bibliography. I suggest reviewing this part more carefully and discuss further. This section should be better presented in order to highlight the most significant and unexpected results, identify correlations, patterns and relationships among the data, speculations, limitations of work and deductive arguments.

Results and discussion section must be considerably improved/ more technically presented. Please apply. All the results obtained should be compared with the reports (preferably recent) in the literature. More care should be taken to present the results and facilitate understanding of the work.

Author Response

Response to Reviewer 2 Comments

Based on this revised version of the manuscript, unfortunately there are still unanswered questions and weaknesses in this paper. Please carefully revise the manuscript according to my previous suggestions.

The discussion presented is poor, in terms of discussing its results and comparing them with the bibliography. I suggest reviewing this part more carefully and discuss further. This section should be better presented in order to highlight the most significant and unexpected results, identify correlations, patterns and relationships among the data, speculations, limitations of work and deductive arguments.

Results and discussion section must be considerably improved/ more technically presented. Please apply. All the results obtained should be compared with the reports (preferably recent) in the literature. More care should be taken to present the results and facilitate understanding of the work.

Response 1: Thank you for taking the time to process our submission entitled "Surface-modified In2O3 for high throughput screening of volatile gas sensors in diesel and gasoline" original paper. Thank you very much for your suggestion. We have revised the manuscript accordingly. As there is very little literature available on gasoline diesel gas sensors, the bibliography listed in the literature comparison is small. To present the advantages of our prepared sensor in a more visual way, we present it in the form of Table 2, to which we have also added a comparison of the response recovery time. The comparison shows visually that our prepared sensor has a lower detection concentration (100 ppm) for diesel and gasoline volatiles, a higher response value and a correspondingly shorter recovery time. We have also modified the results from the discussion to highlight the experimental results more. We have re-capitulated the conclusions in an effort to have better articulation before and after the entire manuscript. These changes we have marked in red.

Round 3

Reviewer 2 Report

This manuscript is now ready for publication in Materials. They have addressed all my concerns.